# Systematic Review of the Role of Alpha-Protein Kinase 1 in Cancer and Cancer-Related Inflammatory Diseases

**DOI:** 10.3390/cancers14184390

**Published:** 2022-09-09

**Authors:** Albert Min-Shan Ko, Hung-Pin Tu, Ying-Chin Ko

**Affiliations:** 1Key Laboratory of Vertebrate Evolution and Human Origins of Chinese Academy of Sciences, Institute of Vertebrate Paleontology and Paleoanthropology, CAS, Beijing 100044, China; 2Department of Public Health and Environmental Medicine, School of Medicine, College of Medicine, Kaohsiung Medical University, Kaohsiung 80756, Taiwan; 3Department of Medical Research, China Medical University Hospital, China Medical University, Taichung 40447, Taiwan; 4Graduate Institute of Toxicology, College of Medicine, National Taiwan University, Taipei 10051, Taiwan

**Keywords:** ALPK1, chronic kidney disease, diabetes, gout, inflammation, oncogene, phosphorylation

## Abstract

**Simple Summary:**

Aside from the basic phosphorylation function of alpha-kinase 1 (ALPK1), little is known about its major functions. Researchers have used various forms of biotechnology and human, animal, and cellular models to better understand the relationship of ALPK1 with cancer and cancer-related inflammatory diseases. ALPK1 is involved in the progression of breast, lung, colorectal, oral, and skin cancer as well as lymphoblastic leukemia. ALPK1 has also been implicated in gout, diabetes, and chronic kidney disease, which are thought to be associated with breast, lung, colorectal, urinary tract, pancreatic, and endometrial cancers and lymphoblastic leukemia. ALPK1 upregulates inflammatory cytokines and chemokines during carcinogenesis. The major cytokine involved in carcinogenesis is TNF-α, which activates the NF-κB pathway, and similar inflammatory responses exist in gout, diabetes, and chronic kidney disease. ALPK1 regulates downstream inflammatory mechanisms that lead to cancer development through certain pathways and plays a key role in cancer initiation and metastasis.

**Abstract:**

Background: Deregulation of conventional protein kinases is associated with the growth and development of cancer cells. Alpha-kinase 1 (ALPK1) belongs to a newly discovered family of serine/threonine protein kinases with no sequence homology to conventional protein kinases, and its function in cancer is poorly understood. Methods: In this systematic review, we searched for and analyzed studies linking ALPK1 to cancer development and progression. Results: Based on the current evidence obtained using human, animal, cellular, and tissue models, ALPK1 is located upstream and triggers cancer cell development and metastasis by regulating the inflammatory response through phosphorylation. Its mRNA and protein levels were found to correlate with advanced tumor size and lymph node metastasis, which occur from the cellular cytoplasm into the nucleus. ALPK1 is also strongly associated with gout, chronic kidney disease, and diabetes, which are considered as inflammatory diseases and associated with cancer. Conclusion: ALPK1 is an oncogene involved in carcinogenesis. Chronic inflammation is the common regulatory mechanism between cancer and these diseases. Future research should focus on identifying inhibitors of serine/threonine and ALPK1 at their phosphorylation sites, which would block various signal transductions and potentially offer kinase-targeted therapeutic agents for patients with cancer and inflammatory diseases.

## 1. Introduction

The human genome contains approximately 538 protein kinase genes, which represent approximately 2% of all human genes [1]. The two main types of protein kinase are serine/threonine kinases and tyrosine kinases. The basic function of protein kinases is phosphorylation, a process by which phosphates are added to target proteins, resulting in functional changes in signal transduction substrates. Relatively few studies have focused on alpha-protein kinase. Early studies reported that the properties of alpha-kinases include phosphorylation of serine/threonine residues on alpha-helix-containing protein substrates [2], and they were shown to phosphorylate Dictyostelium myosin heavy chain A kinase [3]. Ryazanov et al. [4] revealed that the newly discovered alpha-protein kinase has no sequence homology to conventional protein kinases. Later, alpha-kinase 1 (ALPK1) was detected with a molecular weight of 139 kDa and was associated with protein sorting in apical transport vesicles or the trans-Golgi network, where it phosphorylates myosin I [5], indicating that cytoplasmic transport to the apical plasma membrane in epithelial cells plays a key role in exocytic transport. However, the protein structure of ALPK1 is not well understood without X-ray crystallography analysis. The PSIPRED protein sequence analysis workbench [6] predicted the conserved ALPK1 secondary structure and possible function domains and predicted binding sites of protein [7]. ALPK1 is the full length of 1244 amino acids with 16 exons. The N-terminal region contains several amphipathic helices, while the C-terminal region contains an alpha-kinase catalytic domain. Full-length and truncated forms of N-terminal and C-terminal regions are constructed for further studies. 

Following the genome-wide linkage of gout patients on Chromosome 4q25 [8], Wang et al. [9] used microarray mining, cell models, and studies in gout patients to demonstrate that ALPK1 mRNA expression was higher in gout patients than in controls and to establish its involvement in the vesicular transport of certain cytokines to the monocyte plasma membrane. ALPK1 expression was correlated with proinflammatory cytokines, such as IL-1β, TNF-α, and IL-8, with or without monosodium urate. 

The Yamada research team conducted a series of population-based cohort studies [10,11,12,13] in Inabe, Mie, Japan, to study the association between genetic susceptibility and chronic diseases by conducting a genome-wide or genotyping analysis. The rs2074379 (M732I) and rs2074388 (G565D) of ALPK1 variants were found to be significantly associated with type 2 diabetes, whereas rs2074380 (G870A) and rs2074381 (A916G) were significantly associated with chronic kidney disease and myocardial infarction. However, association analysis did not detect a significant association between rs11726117 (M861T) ALPK1 variant and gout susceptibility in the Japanese population. These epidemiological studies did not measure proinflammatory factors, but diabetes, chronic disease, and gout are inflammatory diseases and may be associated with cancer. 

Gout is a common inflammatory disease caused by hyperuricemia attributed to disordered purine metabolism. In a large Swedish study, patients with gout had 25% higher risk of developing cancer and associated malignancies of the oropharynx, colon, liver, pancreas, lung, endometrium, kidney, and skin than individuals without gout [14]. Chronic kidney disease is a risk factor for certain types of cancer, such as colorectal cancer [15,16], and several inflammatory mechanisms have been implicated in a link between cancer and chronic kidney disease [17]. In many systematic reviews and meta-analyses, diabetes mellitus has been associated with some site-specific cancer risks, although the results have been inconsistent [18,19,20]. However, inflammatory cytokines have been attributed to increased risk of cancer in patients with diabetes [21]. Gout, chronic kidney disease, and diabetes are considered inflammatory diseases and have been linked to cancer.

ALPK1, a protein kinase, has also been linked to these cancer-related inflammatory diseases, but its association with cancer was not reported until 2016. In the present systematic review, we search for and analyze the latest evidence regarding the role and mechanism of ALPK1 in the link between cancer and cancer-related inflammatory diseases and suggest the use of protein kinase inhibitors for the treatment of patients with cancer. Studies involving human, animal, cellular, and tissue models support the relationship between ALPK1 and cancer and also suggest pathways and mechanisms of carcinogenesis.

## 2. Materials and Methods

### 2.1. Search Strategy

In June 2022, the PubMed NCBI and Web of Science databases were searched for relevant studies in accordance with the updated Preferred Reporting Items for Systematic Reviews (PRISMA) [22] criteria, and the review was not registered. The following English keywords were used: [ALPK1], [alpha-kinase 1]. Articles retrieved from these databases were published from January 2012 to May 2022. All articles were downloaded to Endnote software X9, filtered by title, and then reviewed along with the abstract. Filters in PubMed NCBI and Web of Science were used to remove review articles, letters, and comments from the searched articles. 

### 2.2. Study Selection and Data Extraction

As depicted in Figure 1, 127 records were obtained from the initial database search. After removing duplicate records and nontext articles, 72 records remained. Thereafter, their abstracts were reviewed to assess study eligibility; on this basis, 22 articles were excluded and 50 articles were systematically reviewed. An additional 35 studies were excluded because they were studies of bacteria, immunity, repeated follow-up, or genetic diseases. Finally, 15 studies published during the past 10 years met the inclusion criteria. Selected measures included balanced studies of relevant patients, human cells and tissues, and animal models. Explanations and comments are provided for all 15 articles.

## 3. Results

### 3.1. ALPK1 Associated with Cancer Development

In a paired case–control study, Liao et al. [23] performed RT-PCR of 47 colon and lung cancer tissues and their adjacent normal fractions to confirm that downregulation of ALPK1 may promote tumorigenesis through mRNA expression. Using cancer cell lines through a healing assay, they discovered that knockdown and overexpression of ALPK1 had no effect on the viability of colorectal and that lung cancer cells did affect cell migration, which plays a major role in the metastasis phase of tumor progression. In addition, through high-resolution melting analysis, the authors found five missense mutations, two frameshifts, and one synonymous mutation of ALPK1 in the C-terminal region that may be associated with colorectal cancer, including three missense mutations, one frameshift, and one synonymous mutation associated with lung cancer. 

By assessing the effect of knockdown of 420 kinases in bilineage triple-negative breast cancer cells expressing myoepithelial and luminal cell characteristics, Strietz et al. [24] revealed that restricting the expression of ERN1 and ALPK1 independently resulted in anchorage-independent loss of growth and reduced tumorigenicity. In vivo, ERN1- or ALPK1-deficient breast cancer cells were found to be less tumorigenic than those without these deficiencies. The gene expression patterns of shERN1 and shALPK1 cells overlapped significantly, suggesting their roles in the same pathway.

Li et al. [25] downloaded adult acute lymphocytic leukemia patient data from the TCGA database and then screened them for differentially expressed genes between samples with good and poor prognosis. Through cluster analysis, 59 patients were determined to have good prognosis (survival time longer than 3 months) and 114 patients were determined to have poor prognosis (survival time shorter than 3 months). After 70 months of follow-up, ALPK1, ACTN4, CALR, and ZNF695 were identified as potential prognostic risk factors for adult acute lymphocytic leukemia in the overall survival analysis.

Ji et al. [26] downloaded a gene expression dataset from the TCGA database that comprised 59 samples from patients with nonrelapsed acute lymphoblastic leukemia (good group) and 114 samples from patients who died of relapse (poor group). Differentially expressed genes were identified between the good and poor groups, and pathway and functional enrichment analyses were performed. ALPK1, ACTN4, CALR, ZNF695, and FBXL5 were identified as novel prognostic genes in relapsed acute lymphoblastic leukemia.

Rashid et al. [27] presented genomic analysis of 75 samples from 57 representative patients, including 15 cylindromas, 17 spiradenomas, 2 cylindroma–spiradenoma hybrid tumors, and 24 spiradenocarcinoma. Using whole-exome sequencing, the authors identified a recurrent missense mutation (V1092A) in the C-terminal kinase domain of ALPK1 gene in the spiradenomas and spiradenocarcinomas; this mutation activated the NF-κβ pathway in a reporter assay.

Chen et al. [28] investigated the expression pattern of ALPK1 in 39 matched patients with oral squamous cell carcinoma and found that the expression of ALPK1 was significantly higher in cancerous tissues than in noncancerous tissues. Moreover, Ki67 staining revealed stronger expression in the nucleus. ALPK1 expression was abnormal in malignant oral cancer cell lines compared to precancerous oral epithelial cells and normal oral epithelial cells. Knockdown of ALPK1 resulted in significantly reduced cell growth, proliferation, migration, and invasion and suppressed the expression of proteins related to epithelial–mesenchymal transition. TNF-α was decreased in ALPK1-depleted metastatic oral cancer cells. 

Lee et al. [29] screened 83 long noncoding RNA arrays stimulated with monosodium urate in monocytes and suggested that HAR1A interacted with ALPK1. In the nucleus of cancer cells, HAR1A functioned upstream of the signaling pathway, and knockdown of HAR1A promoted ALPK1 expression and downregulated myosin IIA and BRD7, leading to inflammation and oral cancer progression. In oral cancer cells and monocytes, the expression of TNF-α and CCL2 was increased after HAR1A knockdown and decreased after ALPK1 knockdown. HAR1A knockdown upregulated the expression of ALPK1 and proteins related to epithelial–mesenchymal transition.

*Fusobacterium nucleatum* has been reported to increase the proliferation and invasiveness of colorectal cells. Zhang et al. [30] observed that *F. nucleatum* induced a novel pattern recognition receptor ALPK1 to activate the NF-κB pathway, resulting in upregulation of intercellular adhesion molecule 1 (ICAM1). In 98 matched colorectal patients, the abundance of *F. nucleatum* in the tumor tissues of patients with colorectal cancer was correlated with the expression levels of ALPK1 and ICAM1, which were also associated with shorter overall survival time in patients with colorectal cancer liver metastasis.

### 3.2. ALPK1 Associated with Gout, Chronic Kidney Disease, and Diabetes

In an epidemiological study, Ko et al. [31] recruited two ethnic groups to study the association of ALPK1 with gout, namely, Taiwanese aborigines (Austronesian people) and Han Taiwanese, comprising 511 gout cases with 840 controls and 104 gout cases with 407 controls, respectively. Using logistic regression models adjusted for age, gender, body mass index, hypertension, hyperuricemia, cholesterol level, triglyceride level, creatinine level, and alcohol use, they discovered that several variants of ALPK1 were associated with gout risk in independent and pooled analyses. One of the ALPK1 missense variants was rs2074388 (G565D) in exon 11, which has also been found in colon cancer [23], chronic kidney disease [32], and diabetes [33]. A NF-κB signal peak upstream of the ALPK1 transcription start site was shown to be present in the lymphoblastoid cell lines.

In blood monocyte assays, Lee et al. [34] found that patients with gout expressed higher levels of ALPK1, myosin IIA, and TNF-α than controls. High-dose colchicine drugs reduced myosin IIA expression but had no effect on ALPK1 expression. The authors stated that monosodium urate stimulated ALPK1 overexpression in monocytes and bonded to myosin IIA through the N-terminus, whereas the C-terminus phosphorylated motor proteins in the presence of ATP. Myosin IIA was activated at the Golgi membrane and transported Golgi-derived TNF-α vesicles toward the plasma membrane, promoting TNF-α secretion through curvature fusion and leading to inflammation in gout flares. 

A cross-sectional and observational study recruited 36 patients with chronic or acute gout and 52 controls from Mexico. ABCG2, SLC22A12, IL-1β, and ALPK1 gene expression was assessed using quantitative real-time PCR. Natsuko et al. [35] reported that ALPK1 expression was positively correlated with serum creatinine level, uric acid (blood monocyte) level, C-reactive protein (blood leukocyte) level, and IL-1β in gout patients, but there was no significant correlation between ALPK1 expression and these biochemical parameters and IL-1β in the total population (patients and controls). 

Employing generalized estimating equation analysis, Yamada et al. [32] conducted a 5-year follow-up genetic study of chronic kidney disease and found that the rs2074379 (M732I) and rs2074388 (G565D) ALPK1 variants were significantly associated with chronic kidney disease and serum concentrations of creatinine with a dominant model. A total of 6027 community residents were recruited from Inabe, Mie, Japan. Chronic kidney disease was diagnosed if the estimated glomerular filtration rate was <60 mL/min/1.73 m^2^. On the basis of this criterion, 655 subjects with chronic kidney disease and 1457 controls were examined and analyzed. 

The same research group [33] found that the rs2074379 (M732I) and rs2074388 (G565D) ALPK1 variants were significantly associated with the prevalence of type 2 diabetes by conducting a longitudinal analysis using generalized estimating equations and adjusting for age, gender, and body mass index in 797 subjects with type 2 diabetes and 5230 controls. These variants of ALPK1 were also associated with fasting plasma glucose level and blood hemoglobin A1c level in all individuals and those not taking antidiabetic medication. Therefore, ALPK1 may be a gene indicating susceptibility to type 2 diabetes mellitus.

Kidney damage is a common result of chronic kidney disease, diabetes, and gout. Kuo et al. [36] investigated the effect of ALPK1 on the development of kidney damage in the context of hyperglycemia. Hyperglycemia was induced in wild-type and ALPK1 transgenic mice by intraperitoneal injection of streptozotocin. Serum glucose concentrations in hALPK1 transgenic mice were higher than in treated wild-type mice, indicating that serum insulin in hALPK1 transgenic mice was reduced. hALPK1 transgenic mice were generated by inducing five missense variants (exons 11 and 13) of human ALPK1 in wild-type mice, and they displayed the same function as human ALPK1. The increase in ALPK1, mainly in diabetic animal and human kidney cell models, leads to accelerated fibrotic nephropathy by enhancing the production of renin, TGF-β1, and IL-1β.

One study [37] identified mediators contributing to the effects of ALPK1 in the induction of nephropathy. The kidneys of streptozotocin-treated ALPK1 transgenic mice were found to express higher levels of CCL2, CCL5, and G-CSF compared to those of controls. Glucose increases ALPK1 mRNA and protein expression in human monocytes and kidney cells. Protein expression of ALPK1, NF-κB, and lectin was upregulated in glucose-treated human kidney cells. ALPK1 was therefore determined to be a mediator involved in diabetic-nephropathy-induced upregulation of CCL2 and CCL5 chemokines.

## 4. Discussion

### 4.1. ALPK1 Is Associated with Cancer Development and Metastasis

Until 2016, no data supported the association of ALPK1 with cancer. Since then, eight studies have obtained evidence supporting the association of ALPK1 with cancer through human, animal, and cellular models, as indicated in Table 1.

#### 4.1.1. ALPK1 and Cancer in Human Models 

Of the four paired case–control studies included herein, three studies revealed that ALPK1 is strongly associated with cancer progression and metastasis, whereas one study demonstrated that ALPK1 is a driver of low- and high-grade spiradenocarcinoma.

A pilot study [23] showed that mRNA overexpression of ALPK1 affected the migration of lung and colorectal cancer cells, but this finding was inconsistent between cancer cell lines and cancer tissues from clinical patients. The authors described a two-stage carcinogenesis model indicating that the expression of ALPK1 differs in the initial and late stages of tumorigenesis. However, a missense mutation, rs2074388 (G565D), at the exon 11 of ALPK1 in colon cancer tissues [23] has also been identified in gout [31], chronic kidney disease [32], and diabetes [33]. In an oral cancer study, Chen et al. [28] observed ALPK1 protein overexpression in advanced oral cancer, suggesting that ALPK1 plays a key role in cancer metastasis.

Rashid et al. [27] demonstrated that ALPK1 mutation (V1092A) at the C-terminus in spiradenoma and spiradenocarcinoma can activate NF-κB signaling in cell reporter systems. Zhang et al. [30] also identified that *F. nucleatum* induces ALPK1 to activate the NF-κB pathway, resulting in upregulation of ICAM1 in two colorectal cancer lines. ALPK1-deficient oral squamous cell carcinoma metastasis leads to reduced TNF-α production in cancer cells [28]. These results suggest that the mechanism by which ALPK1 acts is related to cancer inflammation. 

Li and Ji [25,26] identified ALPK1, together with other genes, as a prognostic risk factor for leukemia. The two studies used the same database and the same research method and obtained almost the same results. It is unusual for the latter paper not to cite and discuss a nearly similar previous paper. Nonetheless, these studies also provide evidence supporting a relationship between ALPK1 and cancer.

#### 4.1.2. ALPK1 and Cancer in Experimental Cell Models

Strietz [24] noted that triple-negative breast cancer cells lacking ALPK1 or ERN1 were less tumorigenic than those not lacking them, suggesting their role in the same pathway. ERN1 is a transmembrane serine/threonine protein kinase for endoplasmic reticulum hemostasis [38], and its downstream target XBP1 is activated for tumor growth [39]. K252a is a serine/threonine protein kinase inhibitor (IC50s of 10 to 30 nM) that has been shown to block neuronal differentiation in rat pheochromocytoma cells and is also a potent inhibitor (IC50 of 3 nM) of tyrosine protein kinase activity of the nerve growth factor trk oncogene [40]. The authors demonstrated that K252a (40 nM) inhibits ERN1 activity and that a dose of 1 μM reduces the colony forming ability of various breast cancer cell lines by 20–50%. Both ERN1 and ALPK1 are serine/threonine protein kinases, but there is no data showing the inhibitory effect of K252a on ALPK1, which needs further study.

Lee et al. [29] revealed that ALPK1 gene expression is highly increased in oral cancer cells compared to dysplastic oral keratinocytes, particularly in the nucleus. In Chen’s qualitative study [28], ALPK1 was rarely expressed in noncancerous tissues but was present in the keratinocyte layer and was strongly expressed in oral cancer tissues. According to further quantitative study by Lee et al. [29], the ALPK1 expression was 26% in dysplastic oral keratinocyte nuclei and increased to 80% in oral cancer cells nuclei. Understanding the mechanisms by which ALPK1 and proinflammatory cytokines enter the nucleus may lead to the development of drugs that prevent cancer progression and metastasis. 

Overall, four studies used samples from patients with cancer and matched them with adjacent normal human tissues as well as appropriate cancer and normal cell lines. These paired case–control studies [23,27,28,30] consistently demonstrated that ALPK1 is associated with cancer in humans [25,26] and in cellular models [24,29], providing evidence of an oncogenic role that can be considered as “probable” in accordance with the International Agency for Research on Cancer [41] classification of carcinogenic agents, which evaluates carcinogenicity on the basis of human and animal studies and mechanistic as well as other relevant data. Furthermore, two independent studies of colorectal cancer or oral cancer have been reported. Therefore, we propose that ALPK1 is an oncogene, although further evidence is needed to verify the findings.

### 4.2. ALPK1 Is Associated with Gout, Chronic Kidney Disease, and Diabetes

Seven recent studies and one early pioneer study have linked ALPK1 to gout, chronic kidney disease, and diabetes, as summarized in Table 2.

#### 4.2.1. ALPK1 and Gout

Human genomic studies have identified many urate transporter genes, including ABCG2, SLC22A12, SLC2A9, and OAT4 as risk factors for gout by conducting genome-wide analyses and meta-analyses [42,43,44]. Ko et al. [30] revealed ALPK1 as a key candidate gene for gout with NF-kB expression in two unrelated ethnic groups. A genomic study [45] analyzed the epistasis of ALPK1 to the urate transporter genes loci and found a positive predictive value (>80%) for gout risk, supporting the role of ALPK1 in causing gout. Correlation analysis of gene mRNA expression was performed using biochemical parameters in blood monocytes and leukocytes and by considering the clinical history of patients with gout. Natsuko [34] revealed that ALPK1 mRNA expression is associated with hypertension and creatinine, uric acid (hyperuricemia), and C-reactive protein levels, also supporting the role of ALPK1 in triggering gout. IL-1β was found to be significantly increased in patient monocytes compared with leukocytes. This was consistent with previous findings [9] demonstrating IL-1β expression in patients with gout. 

According to a study by Lee et al. [34], ALPK1 binds to myosin IIA through the N-terminus, while C-terminus phosphorylates motor proteins in the presence of ATP, transports to the plasma membrane, and contributes to secretion of TNF-α, suggesting ALPK1 is not a transmembrane protein. The properties of curvature membrane fusion must be considered in drug development. Uric acid does not increase the protein expression of ALPK1 but monosodium urate does, suggesting that monosodium urates are endogenous toxic substances in humans. Therefore, we speculate that the pathogenesis of gout involves hyperuricemia leading to precipitation of monosodium urate in joints, which triggers ALPK1 to activate proinflammatory cytokine (such as TNF-α) secretion. This results in inflammation and very painful attacks, thereby causing acute gout flares. Over time, chronic gout can develop without proper control, even into malignant gout.

#### 4.2.2. ALPK1 and Chronic Kidney Disease and Diabetes

The missense variants rs2074379 (M732I) and 2074388 (G565D) of ALPK1 were found to be significantly associated with the prevalence of chronic kidney disease [32] and diabetes [33] in a dominant model, suggesting that the two diseases are closely related. Both variants were also revealed to be associated with increased serum creatinine concentration in chronic kidney disease. Creatinine level was correlated with ALPK1, consistent with findings from studies of patients with gout, as reported by Lee et al. [34] and Natsuko et al. [35], suggesting that creatinine is affected by ALPK1 in chronic kidney disease and gout. 

Findings [36,37] from experimental models of hyperglycemia suggest that TGF-β1, IL-1β, renin, and ALPK1 levels in the kidney or blood is associated with ALPK1-induced fibrotic kidney injury. ALPK1 is a mediator of CCL2 and CCL5 chemokine, and upregulation of NF-κβ is associated with the induction of diabetic nephropathy. These studies of inflammatory mechanisms further support the relationship between ALPK1 and chronic kidney disease and diabetes. 

Numerous studies have supported the finding that ALPK1 is correlated with gout, chronic kidney disease, and diabetes using epidemiological, human, animal, and cellular models. An exception is a study [13] in which only one variant (rs11726117; M861T) of ALPK1 was used to investigate the link between ALPK1 and gout. However, the ALPK1 missense variants rs2074379 (M732I) and rs2074388 (G565D) have been reported to be significantly associated with gout [31], chronic kidney disease [32], and diabetes [33], while the rs2074388 (G565D) variant has also been associated with colon cancer [23], suggesting that ALPK1 is closely related to chronic kidney disease and diabetes.

### 4.3. Gout, Chronic Kidney Disease, and Diabetes Increase Cancer Risk as Shown by Meta-Analysis of Epidemiological Studies 

Although epidemiological and mechanistic studies have implicated ALPK1 in gout, chronic kidney disease, and diabetes, no study has reported whether ALPK1 is directly involved in these inflammatory diseases that contribute to cancer risk. By contrast, the relationship of gout, chronic kidney disease, and diabetes with cancer risk has been investigated in numerous studies with consistent or inconsistent results. Meta-analysis results are considered a more reliable source of evidence than those from a single study because they combine the results of multiple scientifically eligible studies by calculating the pooled relative risk. This review selected only recent large-scale meta-analysis articles examining these inflammatory diseases associated with cancer incidence, as summarized in Table 3 and discussed below. 

Wang et al. [46] analyzed three prospective cohort studies involving 50,358 patients by performing a meta-analysis and found that patients with gout had increased risk of cancer, especially cancers of the urological system, digestive system, and lungs. Similarly, Xie et al. [47] analyzed six prospective cohort studies involving 2,265,083 patients in a meta-analysis and discovered that patients with gout had increased risk of cancer, particularly cancers of the urinary system, digestive system, lungs, and lymphatic/hematological systems. A significant association was also discovered between hyperuricemia and the risk of digestive and urinary tract cancers in men. 

In an analysis of six prospective studies involving 32,057 chronic kidney disease patients, the risk of urinary tract cancer was found to be excessive, but the risk of prostate cancer was low [48]. Komaki et al. [49] assessed the risk of colorectal cancer in patients with chronic kidney disease across 54 studies involving 1,208,767 patients and demonstrated that the risk of chronic kidney disease was significantly increased for colorectal cancer, regardless of transplant history. 

Whether ALPK1 is involved in diabetes and hyperglycemia is a fascinating topic. Numerous studies have reported associations of nearly 20 cancer sites with diabetes by performing a single or meta-analysis. Tsilidis et al. [50] reanalyzed published meta-analysis samples using stricter criteria with more confounding variables excluded and revealed that type 2 diabetes was associated with the incidence of breast, intrahepatic cholangiocarcinoma, colorectal, and endometrial cancers. Causal relationships between type 2 diabetes and the incidence of liver, pancreatic, and endometrial cancers were strongly suggested in the meta-analysis with bias analysis [51]. 

In short, although the results of meta-analyses have demonstrated that gout, chronic kidney disease, and type 2 diabetes are associated with the same or different cancer sites, the results all generally support the relationship between these inflammatory diseases and cancer.

### 4.4. Possible Mechanisms and Pathways of ALPK1-Related Cancer Development and Metastasis 

Because ALPK1 is associated with cancer development, the primary regulatory mechanism is activation of the NF-κB pathway through the inflammatory cytokine TNF-α. The possible mechanisms and pathways of ALPK1-related cancer initiation and metastasis are depicted in Figure 2 and are as follows. TNF-α production in dysplastic, primary, and metastatic ALPK1-depleted human oral (pre)cancer cells is reduced [27,28]. ALPK1 activation of NF-κB leads to spiradenomas or colorectal cancer [27,30]. TNF-α is a major mediator of inflammation and is involved in cancer initiation, progression, invasion, and metastasis. Recent replication studies have demonstrated that TNF-α is associated with breast [52,53], ovarian [51,54], and oral [28,55] cancers through the NF-κB pathway. More data on ALPK1 and TNF-α as well as NF-κB are needed in the future. 

On the basis of an analysis of Madin–Darby canine kidney cells, Heine and colleagues [5] used protein–protein interaction and immunoprecipitation assays to propose that myosin I is phosphorylated by ALPK1, whereas myosin IIA is phosphorylated by ALPK1 [34]. Mazzolini et al. [56] demonstrated that myosin I(a) can act as a tumor suppressor in intestine cancer. By contrast, whether myosin IIA acts as a tumor suppressor or an oncogene remains unclear [57]. Myosin IIA is considered a tumor-suppressing gene [58,59,60], but recent studies have reported that high protein expression is associated with poor overall survival in colorectal cancer [61,62], suggesting that myosin IIA may have an oncogenic role. In this systematic review, ALPK1 was discovered to upregulate myosin IIA in monosodium-urate-treated cells [34], whereas ALPK1 was found to downregulate myosin IIA in oral cancer cells [29]. Different risk factors or carcinogens may lead to different pathways or translocation of ALPK1 into nuclei, thus altering the regulation of cancer progression [27,28]. NF-κB is activated and enters the nucleus to bind to the target genes [63]. The dynamic balance of cancer-related genes between the cytoplasm and nucleus remains to be investigated.

### 4.5. Interrelationships of ALPK1 with Gout, Chronic Kidney Disease, Diabetes, and Cancer

ALPK1 is associated with cancer, gout, chronic kidney disease, and diabetes in humans, animals, and cellular models; they also have common missense variants and higher protein and mRNA expression of ALPK1, suggesting a strong relationship. Different major risk factors may contribute to these diseases, but gout cases are positively associated with diabetes [64] and chronic kidney disease [65], although diabetes is associated with a lower incidence of gout. This suggests that there is a mutual relationship between them. They also share inflammatory mechanisms with proinflammatory cytokines, particularly TNF-α and activation of NF-κB induced by ALPK1 in gout patients, while NF-κB expression is present in diabetic nephropathy hALPK1 mice (TNF-α not detected). Therefore, these diseases are considered to be cancer-related inflammatory diseases. 

Currently, no study has examined the role of ALPK1 in the carcinogenesis of gout, chronic kidney disease, and diabetes, but persistent chronic inflammation may be involved in the carcinogenesis of these diseases. A number of factors have been attributed to increased risk of cancer in people with diabetes [21], and inflammation is a key factor in the link between diabetes and cancer [66]. Chronic inflammation increases production of reactive oxygen species (ROS) and proinflammatory and NF-κB pathway activation [67] and promotes tumor development [68]. Likewise, several ROS and inflammatory mechanisms have been implicated in the link between cancer and chronic kidney disease [17]. The inflammatory mechanisms of gout-related cancer is unclear, but uric-acid-induced ROS plays an important role in carcinogenesis [69], suggesting that there may be a similar persistent inflammatory pathway (Figure 3).

### 4.6. Clinical Implications of ALPK1 and the Development of Preventative and Therapeutic Drugs 

Protein phosphorylation typically exists at multiple sites in a given protein and can alter the function of the protein. Purifying tyrosine-phosphorylated proteins is relatively easy using antibodies, and the phosphorylation sites are relatively well known. Receptor tyrosine kinases are a key family of cell surface receptors involved in extracellular signal transduction. Tyrosine kinase inhibitors have been proven to be effective in cancer therapy; however, serine/threonine kinase inhibitors are less used [70], possibly because they have fewer extracellular signals. The protein structure of ALPK1, which has no sequence homology to conventional protein kinases, is not fully understood, and the purity for commercial use is not ideal and must be strengthened. The phosphorylation sites of ALPK1 and myosin IIA require further exploration, and other substrates may also exist. However, the literature indicates that ALPK1 inhibitors and monoclonal antibodies are potential kinase-targeted therapies and anti-inflammatory drugs for preventing cancer progression and recurrence.

## 5. Conclusions

The literature concerning ALPK1 has been limited to examining the association of ALPK1 with cancer. We divided the related articles into the following three categories: (1) evidence that ALPK1 is associated with cancer; (2) evidence that ALPK1 is associated with gout, chronic kidney disease, and diabetes; and (3) evidence that gout, chronic kidney disease, and diabetes are associated with cancer. Therefore, we could clarify the interrelationships among these disorders on the basis of the epidemiological studies; human, animal, and cellular models; and mechanistic methods. Moreover, colon cancer, gout, chronic kidney disease, and diabetes share common missense variants of ALPK1, indicating that these diseases are closely related.

ALPK1-associated cancers include lung cancer, colorectal cancer, breast cancer, oral cancer, and lymphoblastic leukemia. High ALPK1 expression occurs in precancerous stages, such as oral cell dysplasia, and in skin lesions and persists in both early and advanced cancers, suggesting that ALPK1 is involved in many cancer stages from initiation to metastasis. In cancer and inflammatory diseases, the major inflammasome is TNF-α, which activates the NF-κB pathway, whereas the minor inflammasomes are IL-1β, CCL2, and CCL5. The phosphorylation substrate of ALPK1 is mainly myosin IIA, which is dysregulated in cancer and upregulated in inflammatory diseases, possibly having different phosphorylation sites. Different carcinogens or risk factors may also contribute to differing regulatory mechanisms and pathways. ALPK1 also acts differently in cytoplasm and nuclei. Therefore, we speculate that ALPK1 is most likely an oncogene that phosphorylates substrates and upregulates inflammatory mechanisms, thus contributing to cancer development. Further research is needed to provide evidence of this association and to aid in the development of preventive and therapeutic drugs.

## Figures and Tables

**Figure 1 cancers-14-04390-f001:**
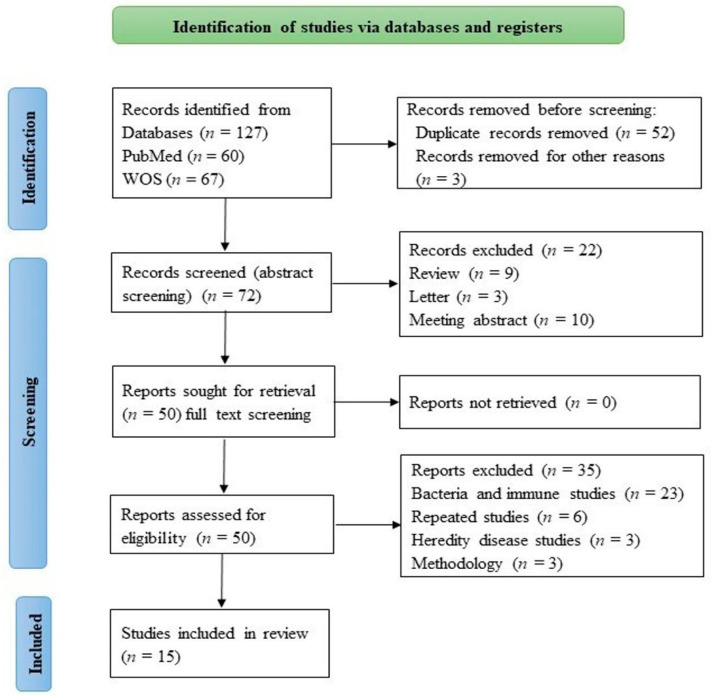
PRISMA flowchart on studies for systematic review.

**Figure 2 cancers-14-04390-f002:**
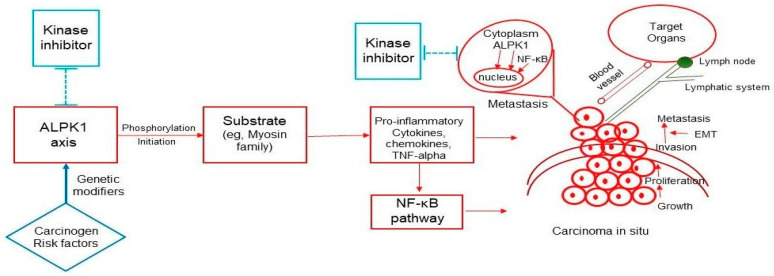
The mechanisms and pathways of ALPK1-related cancer development and metastasis. TNF-α and NF-κβ are the primary mechanisms through which the novel ALPK1 pathway contributes to cancer. In the initial stage, ALPK1 acts upstream to regulate TNF-α, which then activates NF-κB-associated cancers. During the progression stage, a novel pathway involves ALPK1 translocation to the nucleus to promote cancer cell invasion and metastasis through the epithelial–mesenchymal transition (EMT).

**Figure 3 cancers-14-04390-f003:**
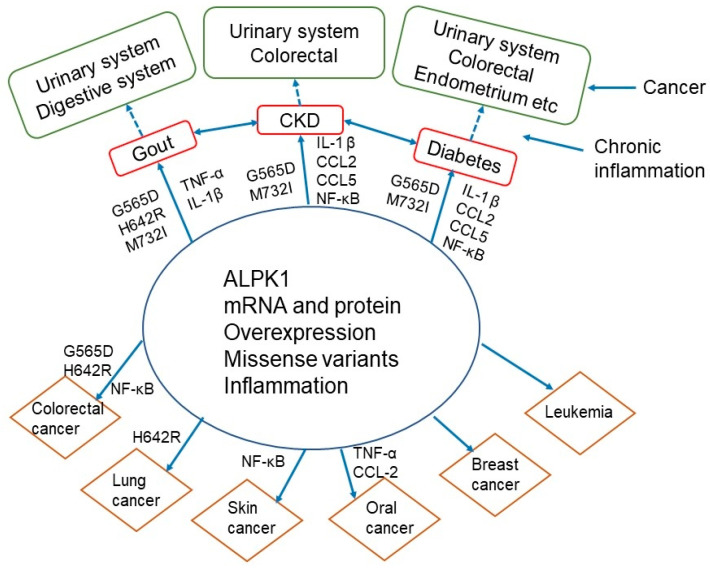
ALPK1 is associated with cancer, gout, chronic kidney disease and diabetes. *ALPK1* variants and inflammatory markers were not detected in all studies. However, they have common missense variants (such as G565D), higher mRNA and protein expression of ALPK1, and also share inflammatory mechanisms. These diseases are considered to be cancer-related inflammatory diseases. Currently, no studies have examined the role of ALPK1 in the carcinogenesis of gout, chronic kidney disease and diabetes.

**Table 1 cancers-14-04390-t001:** Evidence of the association between ALPK1 and cancer.

Author (Reference)	Study Design	Sample	Association/Mechanism
Liao et al. 2016 [23]	Paired case–control study; cellular model	A total of 47 colon and lung cancers and adjacent tissue; colon and lines—primary, late stages	A total of 5–8 variants of ALPK1 in colon or lung cancer tissues; ALPK1 upregulated cancer cell migration in the late stage
Chen et al. 2019 [28]	Paired case–control study; cellular model	A total of 39 oral cancers and adjacent tissue; oral (pre)cancer lines—dysplasia, primary, metastatic stage	ALPK1 was associated with cancer metastasis; TNF-α was decreased in metastatic cells with depleted ALPK1
Rashid et al. 2019 [27]	Paired case–control study;	A total of 42–57 spiradenomas and spiradenocarcinoma and adjacent tissues	A missense mutation of ALPK1 was associated with these benign and malignant skin cancers; NF-κB activation
Zhang et al. 2022 [30]	Paired case–control study; cellular model *Fusobacterium nucleatum*	A total of 98 colorectal cancers and adjacent tissues; colorectal cancer cell lines and endothelial cells	*F. nucleatum*—induced ALPK1/NF-κB/ICAM1 axis regulating colorectal cancer metastasis; shorter survival
Li et al. 2017 [25]	Observational study	Acute lymphoblastic leukemia cases; 114 poor prognosis cases and 59 good prognosis cases	A 70-month follow-up; ALPK1 and a cluster gene acted as prognosis risk factors
Ji et al. 2019 [26]	Observational study	Acute lymphoblastic leukemia cases; 114 poor prognosis cases and 59 good prognosis cases	A 70-month follow-up; ALPK1 and a cluster gene acted as prognosis risk factors
Strietz et al. 2016 [24]	Cellular model; inhibitor effect	Metastatic adenocarcinoma of the breast; a tyrosine kinase inhibitor	Restricting the expression of ALPK1 reduced tumorigenicity; kinase inhibitor decreased cancer cell growth
Lee et al. 2021 [29]	Cellular model; mechanical method	Oral (pre)cancer lines—dysplasia, primary stage; human monocytes	ALPK1 expression increased from 26 to 80% in dysplastic oral cell nucleus and oral cancer cell nucleus; TNF-α and CCL2 expression reduced following ALPK1 knockdown

**Table 2 cancers-14-04390-t002:** Evidence of the association between ALPK1 and gout, chronic kidney disease, and diabetes.

Author (Reference)	Study Design	Sample	Association/Mechanism
Wang et al. [9]	Case–control study; cellular model	A total of 23 gout cases and 39 controls for ALPK1 expression; human monocytes and kidney cells	ALPK1 overexpression in patients with gout; ALPK1 knockdown resulted in decreased IL-1β, TNF-α, and IL-8 mRNA expression
Ko et al. [31]	Population-based case–control study; bioinformatics	Gout cases and controls: 511 and 840 Taiwanese and 104 and 407 Han Chinese, respecitvely	ALPK1 variants were related to excess risk in patients with gout; signal peak of NF-κB at ALPK1 transcription initiation site
Lee et al. [34]	Case–control study; cellular model; proteomic	A total of 20 gout cases and 10 controls; human monocytes	Gout patients expressed higher levels of ALPK1, myosin IIA, and plasma TNF-α; ALPK1 phosphorylated myosin IIA and increased TNF-α secretion in MSU-induced monocytes
Natsuko et al. [35]	Cross-sectional and observational study	A total of 36 gout cases and 52 controls; monocytes and leukocytes; Mexican	ALPK1 expression in gout patients was correlated with serum uric acid, creatinine, C-reactive protein, and IL-1β
Yamada et al. [32]	Population-based cohort study	A total of 655 CKD cases and 1457 controls; Japanese	ALPK1 variants were associated with excess risk and with serum creatinine level in patients with CKD
Yamada et al. [33]	Population-based cohort study	A total of 797 diabetes cases and 5230 controls; Japanese	ALPK1 variants were associated with excess risk in patients with diabetes
Kuo et al. [36]	Animal/cellular models	Three groups of mice; wild type; STZ-treated wild type, STZ-treated hALPK1 transgenic mice	ALPK1 accelerated nephropathy in STZ-induced hyperglycemic mice; levels of IL-1β and TGF-β1 were increased in hALPK1 transgenic mice
Lee et al. [37]	Animal/cellular models	Added hALPK1 transgenic mice group; human kidney cell lines, human monocytes	NF-κB, chemokine CCL2 and CCL5 expression was increased in STZ-treated diabetic nephropathy hALPK1 mice; glucose elevated ALPK1 expression in cells

**Table 3 cancers-14-04390-t003:** Meta-analysis of epidemiological studies demonstrating that gout, chronic kidney disease, and diabetes increase cancer risk.

Author (Reference)	Study Design	Participants	Site of Cancer and Incidence Pooled Relative Risk (95% CI)
Wang et al. [46]	Prospective cohort study	Three studies involving 50,358 individuals for gout and cancer	All: 1.42 (1.09–1.84); urology: 1.72 (1.30–2.26); digestive system: 1.39 (1.23–1.56); lung: 1.29 (1.01–1.65)
Xie et al. [47]	Prospective cohort study	Six studies involving 226,083 individuals for gout and cancer	All: 1.19 (1.12–1.25); urinary system: 1.28 (1.11–1.48); digestive system: 1.15 (1.07–1.24); lung: 1.11 (1.01–1.21)
Wong et al. [48]	Population-based cohort study and randomized controlled trial	Six studies involving 32,057 individuals for chronic kidney disease and cancer	Urinary eGFR <45: 1.66 (1.02–2.70); dialysis: 2.34 (1.31–4.18)
Komaki et al. [49]	Retrospective cohort study	A total of 54 studies involving 1,208,767 individuals for chronic kidney disease and cancer	Colorectum without kidney transplantation: 1.18 (1.01–1.37); after kidney transplantation: 1.40 (1.15–1.71)
Tsilidis et al. [50]	Reanalysis of previous meta-analysis with random effect and 95% prediction intervals	A total of 27 meta-analyses involving > 1 million individuals for type 2 diabetes and cancer	Breast: 1.20 (1.12–1.28); colorectum: 1.27 (1.21–1.34); endometrium: 1.97 (1.71–2.27); intrahepatic cholangiocarcinoma: 1.97 (1.57–2.46)
Liang et al. [51]	Reanalysis of previous meta-analysis with bias analysis	151 cohorts comprising 32 million people for type 2 diabetes and cancer	Very likely causal relationship: liver, pancreatic, endometrium (100%); gallbladder (86%); kidney, colon, colorectal system (>60%)

## Data Availability

Not applicable.

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
