# Peer review of "Systematic Review of the Role of Alpha-Protein Kinase 1 in Cancer and Cancer-Related Inflammatory Diseases"

_cancers, 2022, doi:10.3390/cancers14184390_

Round 1

Reviewer 1 Report

Protein phosphorylation is one of the main regulatory factors that induce and maintain a large number of cellular processes. Importantly, dysregulation of most of the kinases is involved in the initiation and maintenance of a variety of diseases.  This study (review) describes the role of alpha-protein kinase 1 (ALPK1) in inflammation-related diseases as well as cancer. The ALPK1 is an a-typical protein kinase with at least eight distinct variants that seem to regulate cytoskeletal elements and production of various cytokines, probably due to its efficient activation of the NFkB pathway. Using a literature search, the authors report that the ALPK1’s mRNA and protein levels correlate with advanced tumor size and lymph node metastasis, ALPK1 is also strongly associated with gout, chronic kidney disease, and diabetes, which are considered inflammatory diseases and associated with cancer. They conclude that ALPK1 is an oncogene, probably due to induction of inflammation, which is also involved in the other diseases. 

Overall, this is an interesting and important study (review) that shed a light on an important protein kinase that is involved in the induction of many diseases. Not enough reviews on this kinase have been published thus far, and therefore, this study is much needed to the field. However, there are some comments that should be addressed in order to make this manuscript ready for publication. These are as follows: 

1)     The introduction is relatively confusing, both in the development of the information and with the English that is not always clear. In many instances I had to read the material several times to understand the topic. I suggest to re-write this section in a more methodological manner.

2)     In continuation with point 1, there are some inaccuracies in the introduction. For example, the authors write about two main types of protein kinases, they mention three, and omit the histidine kinases. The substrates are not always signal transduction ones. Tyrosine kinases are not only receptors. There are plenty of studies on Ser/Thr protein kinases (more than 3,000) and not only a few. ALPK1 is clearly not a MAPK, and MAPKs are themselves Ser/Thr kinases, so this is not a dual specificity. The history of the field is not accurate. It is suggested to use a help by a kinase expert when reshaping these sections. 

3)     The authors mention the mutation and variants of ALPK1. However, it is not clear what are these mutations and sequence changes, and whether any of them is correlated to any of the diseases, and how. 

4)     In continuation with the previous point, it is suggested to include some biochemical and structural characteristic of the main isoform as well as its mutations and variants.

5)     In the discussion, it is suggested to include a summarizing paragraph for each disease, not just description of distinct studies. The summaries that can be found are not always clear.

6)     Fig. 3, although important, is not very clear to me. It is suggested to make them clearer, in particular the role of inflammation in the separate diseases. 

Author Response

Reviewer 1

Comments and Suggestions for Authors

Q1. The introduction is relatively confusing, both in the development of the information and with the English that is not always clear. In many instances, I had to read the material several times to understand the topic. I suggest to re-write this section in a more methodological manner.

A1. Thank you very much. We have rewritten these sentences to be more clear. Please refer to Introduction section.

Q2.   In continuation with point 1, there are some inaccuracies in the introduction. For example, the authors write about two main types of protein kinases, they mention three, and omit the histidine kinases. The substrates are not always signal transduction ones. Tyrosine kinases are not only receptors. There are plenty of studies on Ser/Thr protein kinases (more than 3,000) and not only a few. ALPK1 is clearly not a MAPK, and MAPKs are themselves Ser/Thr kinases, so this is not a dual specificity. The history of the field is not accurate. It is suggested to use a help by a kinase expert when reshaping these sections. 

A2. Thank you for pointing out some inaccuracies. We have removed these sentences. Please refer to Introduction section.

Q3. The authors mention the mutation and variants of ALPK1, However, it is not clear what are these mutations and sequence changes, and whether any of them is correlated to any of the diseases, and how. 

A3. Thank you for your comments and suggestions. We have added these mutations sequence changes in whole text. Please refer to section 4.1.1. “a missense mutation rs207488 (G565D) at the exon 11 of ALPK1 in colon cancer tissues [23] has also been identified in gout [31], chronic kidney disease [32], and diabetes [33], and 4.2,2.

Q4.  In continuation with the previous point, it is suggested to include some biochemical and structural characteristic of the main isoform as well as its mutations and variants.

A4. Thank you very much for your suggestion. The secondary protein structure of ALPK1 is not well understood without x-ray crystallography analysis. We have added currently known characteristic “ALPK1 is a full-length of 1244 amino acid and 14 exons. The N-terminal region (1-569 amino acids) contains several amphipathic helices, while the C-terminal region (569-1244 amino acids) contains an alpha kinase catalytic domain. Please refer to Introduction section.

Q5.  In the discussion, it is suggested to include a summarizing paragraph for each disease, not just description of distinct studies. The summaries that can be found are not always clear.

A5. Thank you very much for your suggestion. We have added short summaries of some sentences for each disease. Please refer to last part of section 4.1., 4.2., and 4.3.

Q6.  Fig. 3, although important, is not very clear to me. It is suggested to make them clearer, in particular the role of inflammation in the separate diseases. 

A6. Thank you very much. We have removed unclear parts and have rewritten a paragraph and statement on Fig. 3.  To be more clear. Please refer to section 4.5.

Reviewer 2 Report

Ko et al performed a systematic review of the involvement of alpha-protein kinase 1 (ALPK1) in cancer and inflammatory diseases, including gout, diabetes, and chronic kidney disease. The authors identified and selected 15 studies investigating the role of ALPK1 in cancer and disease for the systematic review. The authors then summarized the evidence for the involvement of ALPK1 in different types of cancer, in inflammatory diseases (gout, diabetes, and chronic kidney disease), and of the potential links between inflammatory diseases and cancers.

Comments:

1.       The Discussion would need some rewriting as currently several parts of the Discussion are restatement of the Results.

2.       Line 58: There are many studies on serine/threonine protein kinases. Maybe the authors want to say alpha-protein kinases.

3.       Paragraphs 3.1.3 and 3.1.4: Have the two papers used exactly the same patients for their analysis (numbers are the same)? I would be better to precise if there are or not exactly the same patients between both papers.

4.       Paragraph 3.2.3: The lines 216-219 are not clear on what is correlated and what is not significant. Are all the correlation not significant or only some of them, and if some which ones?

5.       Paragraph 4.1.2: The authors should succinctly describe what are the functions of ERN1 as there are currently unclear. Also, the authors should precise what is the target of K252a, the tyrosine protein kinase inhibitor.

6.       Lines 354-356: Unclear. Is it hyperuricemia/monosodium urate that induce gout, which in turn induce TNFa expression or hyperuricemia/monosodium urate that induce gout and TNFa expression?

7.       Lines 477-478: The phrase needs to be rewritten. It is the protein phosphorylation sites and not the protein kinases that exist at multiple sites in a given protein.

Author Response

Reviewer 2

Comments:

Q1. The Discussion would need some rewriting as currently several parts of the Discussion are restatement of the Results.

A2. Thank you very much. We have done our best to remove the sentences of restatement of the Results and have added comments and interpretation on the significance of the articles.  Please refer to section 4.1. and 4.2.

Q2. Line 58: There are many studies on serine/threonine protein kinases. Maybe the authors want to say alpha-protein kinases.

A2. Thank you for pointing this out. We have revised it. Please refer to Introduction section.

Q3. Paragraphs 3.1.3 and 3.1.4: Have the two papers used exactly the same patients for their analysis (numbers are the same)? I would be better to precise if there are or not exactly the same patients between both papers.

A3. Thank you very for your comments. We have no the raw data to judge the two papers, only tell the truth at this time. “It is unusual for the latter paper not to cite and discuss a nearly similar previous paper.” Please refer to section 4.1.1. last part.

Q4. Paragraph 3.2.3: The lines 216-219 are not clear on what is correlated and what is not significant. Are all the correlation not significant or only some of them, and if some which ones?

A4. Thank for your comments. We have rewritten these sentences to be more clear. “but there was no significant correlation between ALPK1 expression and these biochemical parameters and IL-1B, in the total population (patients and controls).” Please refer to section 3.2.3.

Q5. Paragraph 4.1.2: The authors should succinctly describe what are the functions of ERN1 as there are currently unclear. Also, the authors should precise what is the target of K252a, the tyrosine protein kinase inhibitor.

A5. Thank you for your valuable comments. We have cited more ERN1 and K252a studies and have added some sentences to explain the function of ERN1. Also, we have corrected the inhibitory effect of K252a on ERN1. Please refer to section 4.1.2.  

Q6. Lines 354-356: Unclear. Is it hyperuricemia/monosodium urate that induce gout, which in turn induce TNFa expression or hyperuricemia/monosodium urate that induce gout and TNFa expression?

A6. Thank you very much. We have rewritten the pathogenesis of gout. “hyperuricemia leads to the precipitation of monosodium urate in joints, which triggers ALPK1 to activate pro-inflammatory cytokines (such as TNF-a) secretion resulting in inflammation and very painful attacks, thereby causing acute gout flares.”  Please refer to section 4.2.1.

Q7. Lines 477-478: The phrase needs to be rewritten. It is the protein phosphorylation sites and not the protein kinases that exist at multiple sites in a given protein.

A7. Thank you for pointing it out. We have rewritten it. Please refer to section 4.6.

Round 2

Reviewer 1 Report

The paper has been significantly improved. I still have two comments

In the Graphical abstract it seems that ALPK1-induced cancer is always through inflammation. Is there a possibility to show that some cancer can be induced through other mechanisms.

In line 15 of the introduction, the term secondary structure is probably wrong. You can just omit the word secondary. 

Author Response

Reviewer 1 has two comments:

We are grateful to the carefully reading and the useful advice. 

1. In the abstract, is there a possibility to show that some cancer be induced through other mechanisms?

Answer: In the currently reviewed paper, there is no information on other mechanisms, but we agree that there may be potential other mechanisms. Therefore, in the last sentence of the abstract, we have added a word, “various”.

2.In the line 15 of the introduction, you can omit the word secondary.

Answer: We have omitted the word “secondary”.